# Optimization of Electron Transport Layer Inkjet Printing Towards Fully Solution-Processable OLEDs

**DOI:** 10.3390/ma18143231

**Published:** 2025-07-09

**Authors:** Riccardo Manfredi, Carmela Tania Prontera, Fabrizio Mariano, Marco Pugliese, Antonio Maggiore, Alessandra Zizzari, Marco Cinquino, Iolena Tarantini, Giuseppe Gigli, Vincenzo Maiorano

**Affiliations:** 1CNR-NANOTEC-Institute of Nanotechnology, c/o Campus Ecotekne, Via Monteroni, 73100 Lecce, Italy; riccardo.manfredi@nanotec.cnr.it (R.M.); carmelatania.prontera@enea.it (C.T.P.); antonio.maggiore@cnr.it (A.M.); alessandra.zizzari@cnr.it (A.Z.); marco.cinquino@nanotec.cnr.it (M.C.); giuseppe.gigli@cnr.it (G.G.); vincenzo.maiorano@cnr.it (V.M.); 2Laboratory of Hydrogen and New Energy Vectors (TERIN-DEC-H2V) ENEA–C.R Brindisi, S.S.7 Appia, km 713,700, 72100 Brindisi, Italy; 3Department of Mathematics and Physics, University of Salento, Via Monteroni, 73100 Lecce, Italy; iolena.tarantini@unisalento.it

**Keywords:** inkjet printing, ink optimization, electron transport layer, OLEDs

## Abstract

The fabrication of high-performance organic optoelectronic devices using solution-based techniques, in particular inkjet printing, is both a desirable and challenging goal. Organic light-emitting diodes (OLEDs) are multilayer devices that have demonstrated great potential in display applications, with ongoing efforts aimed at extending their use to the lighting sector. A key objective in this context is the reduction in production costs, for which printing techniques offer a promising pathway. The main obstacle to fully printed OLEDs lies in the difficulty of depositing new layers onto pre-existing ones while maintaining high film quality and avoiding damage to the underlying layers. In a bottom-emitting OLED, the electron transport layer (ETL) is the final organic layer to be deposited, making its printing particularly challenging, a process for which only a few successful examples have been reported. In this work, we report on the optimization of a 2,2′,2″-(1,3,5-Benzinetriyl)-tris(1-phenyl-1-H-benzimidazole) (TPBi)-based ink formulation for ETL printing on an emitting layer composed of 5,10-Bis(4-(3,6-di-tert-butyl-9H-carbazol-9-yl)-2,6-dimethylphenyl)-5,10-dihydroboranthrene (tBuCzDBA). A specific ratio of methanol to diethyl ether was identified as the most suitable for printing the ETL without compromising the integrity of the underlying layer. The printed ETL was successfully integrated into an OLED device, which exhibited a maximum current efficiency of 6.8 cd/A and a peak luminance of about 8700 cd/m^2^. These results represent a significant step toward the development of a fully printed OLED architecture.

## 1. Introduction

Physical Vapor Deposition (PVD) is a common OLED fabrication technique, valued for its capability to achieve highly precise deposition of ultra-thin, high-purity films. Despite its widespread industrial application, the reliance on vacuum systems and complex deposition setups significantly increases operational costs. Additionally, the requirement for vacuum chambers and masks to create patterned films hinders the fabrication of large-area optoelectronic devices and limits design versatility. Consequently, solution-based deposition techniques are emerging as promising alternatives due to their scalability and adaptability to flexible and large-area substrates [1,2]. They include dip coating, spin coating, blade coating, slot-die coating, spray coating, screen printing, gravure printing, and inkjet printing [3,4,5,6,7,8]. Among all solution-based techniques, inkjet printing stands out as one of the most interesting, due to its versatility in pattern geometry and its operational simplicity [9,10,11,12,13,14,15]. Inkjet printing technology is based on the ejection of ink droplets to reproduce a digital image and, as such, involves no contact between the substrate and the printer components. As a result, ink consumption is minimal, and print quality can be controlled by modifying the chemical–physical properties of the ink [15,16]. Considering these advantages, inkjet printing is a valid alternative to obtain a fully printed OLED device. To this end, one of the most challenging aspects is optimizing the ink formulation for the printing process on the pre-existing organic layers to achieve a well-defined multilayer structure. The ink must enable the deposition of a uniform, high-quality thin film while preventing damage and/or redissolution of the underlying layer, which could compromise the integrity of the multilayer architecture and the proper functioning of the electro-optical device [15,17,18,19]. In this context, printing on top of organic layers composed of small molecules (as is typical for more efficient and widely used emitting materials) could be particularly challenging since the redissolution effect is more relevant and the search for orthogonal solvents is more difficult [20]. For these reasons, in a typical bottom-anode OLED structure, printing the last organic layer (typically the ETL) onto the emitting layer (EML) remains one of the most critical steps in the fabrication of a fully printed organic structure.

OLEDs with a solution-based ETL on top of the EML have been fabricated using spin coating, slot-die coating, blade coating, and, in a few cases, inkjet printing has also been reported [10,12,13,21,22,23,24,25,26,27,28,29]. TPBi-based ETLs from ethanol solution have been deposited by spin coating on top of a thermally activated delayed fluorescence (TADF) dendrimer [13], as well as a TADF polymer [10]. Alternative ETLs based on oxide semiconductors have also been deposited by spin coating. Nagar et al. [23] reported depositing a TiO_2_-doped TPBi film as an ETL in a phosphorescent OLED, while Oh et al. [24] developed a new solution-processable ETL based on ZnO nanoparticles and cesium carbonate-doped ethoxylated polyethyleneimine, exploiting a polymethyl methacrylate interlayer to enhance wettability. Slot-die coating is another valid alternative for solution-processable OLEDs, and examples have been reported for the deposition of ETLs based on TPBi and ZnO nanoparticles [21,22]. TPBi-based ETL has also been deposited by blade coating (Yeh et al.) [26]. Finally, the few examples of printed ETLs reported in the literature lack detailed information on ink formulation, highlighting the relevance of and need for further research in this area [12,27,28].

In this work, we optimized an inkjet-printable TPBi solution and demonstrated the performance of OLEDs using it as the ETL, deposited on top of an EML based on tBuCzDBA. tBuCzDBA is a high-performing TADF emitter used by Wu et al. [30] in an OLED architecture based on a thermally evaporated host-guest system, and we have recently demonstrated inkjet printing of the same material as a self-hosted emitting layer [11]. To print the ETL on top of the tBuCzDBA-based layer, both the ink formulations and the EML’s thin-film composition were optimized. A mixture of methanol and diethyl ether in a ratio of 8:2 was selected as the optimal solvent composition to dissolve and print the ETL compound, while an emissive layer consisting of a mixture of high molecular weight polyvinylcarbazole (PVK) and tBuCzDBA was chosen to enable ETL printing without damaging the EML. The results reported here will contribute to the future development of a fully printed OLED device.

## 2. Materials and Methods

### 2.1. Chemicals and Reagents

PEDOT:PSS AI 4083 was purchased from Heraeus (Hanau, Germany), and tBuCzDBA was provided by Lumtec (Hsinchu, Taiwan). ITO-covered glass substrates were purchased from Kintec (Hong Kong, China), and aluminum pellets (Al) were provided by Kurt J. Lesker (Jefferson Hills, PA, USA). The following materials were purchased from Sigma-Aldrich (St. Louis, MO, USA) without any further purification: anhydrous chloroform (CF, ≥99%), anhydrous chlorobenzene (CB, ≥99.8%), PVK (average M_w_ ~ 1,100,000, powder), methanol (MeOH, ≥99.8%), diethyl ether (Et_2_O, ≥99%), TPBi, and LiF.

### 2.2. Ink and Film Characterization

The CAM 200 (KSV Instruments Ltd., Helsinki, Finland) instrument was used to perform pendant drop measurements and evaluate the surface tension of all solutions. The reported data are the average of three measurements with standard deviations from the medium value. The same CAM 200 apparatus was used for the static contact angle measurements of all the solutions by the sessile drop method. Several drops of each solution were deposited onto different areas of the selected substrates and observed for 60 s. Two substrate architectures were characterized: glass ITO/PEDOT:PSS/PVK/tBuCzDBA and glass ITO/PEDOT:PSS/PVK/PVK:tBuCzDBA (prepared as reported in Section 2.3, tBuCzDBA thin film was obtained by spin coating a 10 mg/mL solution in chlorobenzene at 4000 rpm for 40 s, followed by thermal treatment at 110 °C for 10 min).

Film thicknesses were measured through a surface profiler (Dektak, Tucson, AZ, USA) characterized by a mechanical stylus with a 12 μm tip diameter. Inkjet-printed films were characterized by Atomic Force Microscopy (Nanosurf EasyScan 2, Binningen, Switzerland) in the non-contact mode using silicon tips with a nominal tip radius being less than 10 nm, and WS × M was used for the image analysis.

Fluorescence spectroscopic studies were performed with an Edinburgh FLS980 spectrometer equipped (Livingston, UK) with a Peltier-cooled Hamamatsu R928 photomultiplier tube (185–850 nm) (Hamamatsu, Japan), at 25 °C. Corrected spectra were obtained via a calibration curve supplied with the instrument. For the emission spectra, the widths of the slits were adjusted between 1 and 5 nm, integration time = 0.1 s, 1 nm step.

### 2.3. OLEDs Fabrication and Characterization

The device architecture is as follows: glass ITO/PEDOT:PSS/PVK/PVK:tBuCzDBA/TPBi/LiF/Al. At first, the ITO glass substrates were cleaned with isopropanol and acetone and treated with O_2_ plasma cleaner. PEDOT:PSS AI4083 solution was spin-coated at 4000 rpm for 40 s, and then the thin film was placed on a hot plate at 140 °C for 10 min. PVK thin film was obtained starting from a 5 mg/mL solution in chlorobenzene/chloroform 5/95 volume ratio by spin coating at 7000 rpm for 60 s, followed by a thermal treatment at 110 °C for 10 min. Finally, PVK:tBuCzDBA (70:30–10 mg/mL in chlorobenzene) was spin-coated at 4000 rpm for 40 s, and the thin film was thermally treated at 110 °C for 10 min.

TPBi layers were printed from a MeOH:Et_2_O/8:2 solution at different concentrations (2, 2.5, and 4 mg/mL) using a custom-made inkjet printer fabricated by T.P.A. s.r.l. (Sesto San Giovanni, Italy), based on an electromagnetically actuated microdispenser with a 100 µm nozzle. Droplets were ejected in drop-on-demand (DoD) mode through the rapid vertical displacement of a microneedle inside a cylindrical chamber filled with the ink. An electrical pulse activates the actuator, generating a localized pressure that expels a single droplet without contact with the substrate. Printing parameters, such as actuation pulse, nozzle–substrate distance, backpressure, and environmental conditions, were optimized to ensure stable droplet formation and uniform film coverage over the target area. The nozzle ejects drops with a volume of ~0.5 nL from a height of 7 mm above the substrate. The individual circular drops are 0.2 mm separated (center to center). After the deposition, the thin film is dried at room temperature for 1 min and then placed on a hot plate at 110 °C for 5 min. Subsequently, 0.8 nm LiF and 100 nm Al were deposited by thermal evaporation in a Kurt J. Lesker multiple high-vacuum chamber system. The effective light-emitting area of fabricated devices is 15 mm^2^. The optoelectronic characteristics of the OLED devices were measured in a glove box using an Optronics OL770 spectrometer coupled to the OL610 telescope unit with an optical fiber for the luminance measurements. The whole system was directly connected by an RS232 cable to a Keithley 2420 current–voltage source meter.

## 3. Results and Discussion

### 3.1. Ink Optimization and Thin Films Characterization

To obtain a suitable ink, a solvent must be capable of dissolving the printable material without damaging the underlying layer. This fundamental requirement is particularly complex to meet in an OLED manufacturing process because most of the organic materials used are soluble in very similar solvents.

TPBi was selected as the electron transport material to be printed onto a bottom-anode OLED, as it is widely used as an ETL in high-performance OLEDs. Recent investigation has ranked TPBi solubility in various solvents, in ascending order, as follows: cyclohexane < n-hexane < isopropanol < acetonitrile < methanol < ethanol < ethyl acetate < n-butanol < dimethyl sulfoxide < tetrahydrofuran < chlorobenzene < trichloromethane [31]. Given the solubility profile, chlorinated solvents could be used to dissolve and print TPBi; however, they also readily dissolve the selected active material, tBuCzDBA. We, therefore, selected an intermediate solvent (methanol, MeOH) as a starting point for optimizing the ink formulation for TPBi printing. In addition, two different types of active layer were tested to assess their resistance to the printing/solvent-based deposition of the ETL. We have tested pure tBuCzDBA and a blend of PVK and tBuCzDBA (PVK:tBuCzDBA). In this context, the interaction between high molecular weight PVK and tBuCzDBA plays a critical role in enhancing the film′s resistance to various solvents. In addition, PVK is a well-known host material that prevents exciton quenching by mitigating the packing of tBuCzDBA molecules [32]. This synergistic relationship could not only preserve the functional properties of the EML, but also significantly improve the overall solvent resistance of the film.

The first tested ink (#1) consists of a MeOH solution of TPBi at a concentration of 2.5 mg/mL, with a surface tension of 25.5 ± 1.1 mN/m. Its contact angle was evaluated on two different OLED stacks: indium tin oxide (ITO)/poly (2,3-dihydrothieno-1,4-dioxin)-poly (styrenesulfonate) (PEDOT:PSS)/PVK/tBuCzDBA and ITO/PEDOT:PSS/PVK/PVK:tBuCzDBA (Table 1, Figure 1a and Figure 2a). In addition, to evaluate the resistance of the underlying layers to the solvent, the effect of a MeOH droplet on the two active layers (two OLED stacks) was examined under UV illumination. The photoluminescence was monitored before and after solvent evaporation to assess any changes induced by the MeOH exposure (Figure 1b and Figure 2b).

As can be seen in Figure 1a, ink #1 has a contact angle on the pure tBuCzDBA-based stack of 53° at 0 s, which decreases to 24° and 11° after 30 and 60 s, respectively. Although the contact angle is significantly lower after 60 s, the high initial value and the long wetting/drying time are not beneficial for the ink printability. A similar behavior is also observed on the PVK:tBuCzDBA-based stack, although the initial contact angle is slightly lower (Figure 2a, Table 1). Concerning the effect of the solvent on the underlying layers, evidence of damage and/or dissolution effect is visible on both types of active layers (Figure 1b and Figure 2b), which precludes the use of the MeOH-based ink for manufacturing the final device.

Observation of the film under UV light after exposure to a MeOH droplet reveals a marked coffee ring and dewetting effect, as the most significant reduction in fluorescence occurs in the central region of the drop, where the solvent remains concentrated during the final stage of the drying process.

Based on ink #1, two additional formulations were prepared by incorporating varying amounts of diethyl ether (Et_2_O) as a co-solvent (Table 1). Diethyl ether was selected to reduce the ink’s surface tension and contact angle, thereby improving wetting on the substrate and accelerating the drying process [14]. Inks #2 and #3 consist of a solvent mixture of MeOH and Et_2_O with a volume ratio of 8:2 and 6:4 and have a surface tension of 23.1 ± 0.8 and 22.1 ± 0.5 mN/m, respectively (Table 1).

Compared to ink #1, ink #2 shows a lower contact angle on both kinds of OLED stack, as expected due to its lower surface tension (Figure 1c and Figure 2c, Table 1). As a result, the time required for the substrate to be wetted is significantly lower. Ink #2 on top of the pure tBuCzDBA-based stack reaches a contact angle of 5° after only 20 s (Figure 1c), while on the PVK:tBuCzDBA-based stack, the contact angle is slightly higher, with a value of 14° after 20 s (Figure 2c). Such a significantly lower contact angle and reduced wetting/drying time are very beneficial to the quality of the printed thin film [14]. Regarding the effect of a drop of the solvent mixture on the uniformity of the underlying film, damage can be seen on the pure tBuCzDBA-based stack (marked coffee ring effect and accumulation of material in the central part of the drop) (Figure 1d), while the PVK:tBuCzDBA-based one does not appear to be affected, as no coffee-ring effect is visible, the halo of the drop drying process is only slightly noticeable, and the fluorescence of the film remains uniform (Figure 2d). Therefore, the image reported in Figure 2d demonstrates that the solvent mixture of ink #2 does not redissolve the active layer based on PVK:tBuCzDBA, and so the fluorescence remains almost unchanged.

When the amount of Et_2_O is further increased, as in the case of ink #3, a peculiar effect on the contact angle values is observed (Figure 1e and Figure 2e, Table 1). In fact, although the surface tension value is lower than that of #2 (Table 1), in accordance with the increased quantity of the solvent showing a lower surface tension, the initial contact angle is higher than that observed for #2, both on pure tBuCzDBA-based and PVK:tBuCzDBA-based stacks (Figure 1e and Figure 2e, Table 1). Consequently, the time required to achieve a significant reduction in contact angle—and thus good substrate wettability—increases compared to ink #2. In particular, the contact angle on pure tBuCzDBA-based stack reaches 11° after 60 s (Figure 1e), while on the PVK:tBuCzDBA-based stack, the contact angle reaches 7° after 60 s (Figure 2e).

The reason for this trend reversal could be found in the higher capability of formulation #3 to redissolve the underlying organic film, as suggested by the images in Figure 1f and Figure 2f. Consequently, the surface modification of the organic film induces a variation in the interfacial tension between the ink and the substrate, with a significant impact on the contact angle. Furthermore, as previously mentioned for ink #1, a high contact angle negatively affects the print quality.

For this reason, ink #3 can be considered unsuitable for the printing process, while ink #2 provides the best results in terms of drying behavior and preservation of the active layer. Moreover, the active layer based on the PVK:tBuCzDBA blend proves to be the most suitable option for ETL printing.

As further confirmation, the photoluminescence spectrum of the best-performing OLED stack based on PVK:tBuCzDBA active layer was recorded before and after the solvent printing using MeOH:Et_2_O 8:2, corresponding to the composition of ink #2, identified as the most effective formulation. The results are shown in Figure 3a,c.

As shown in Figure 3a, solvent printing on the PVK:tBuCzDBA blend results in a decrease in the PL intensity of the emitting layer. This effect, also evident in the image of the PVK:tBuCzDBA-based stack under UV light after solvent printing (Figure 3c), is attributed to the coffee ring effect (namely, a partial redissolution of the active layer followed by material accumulation at the edges of the printed area, resulting in a local thinning of the film) [33]. A reduction in the thickness of the active layer of approximately 10–15% was evaluated through thickness measurements. Nonetheless, this phenomenon does not compromise the overall quality and uniformity of the resulting film.

The morphological characterization of the TPBi film printed with ink #2 on the PVK:tBuCzDBA-based stack confirms the suitability of both the selected ink and active layer for OLED device fabrication (Figure 3b). The AFM image reveals material clusters-zones rich in TPBi only a few nanometres in height (4–5 nm), along with a remarkably low surface roughness (Root Mean Square, RMS, 1.1 nm).

For comparison, the photoluminescence spectrum of the tBuCzDBA-based stack was evaluated after solvent printing, and the surface morphology of the corresponding printed TPBi was analyzed. The results are shown in Appendix A.

These findings confirm that a high-quality TPBi-based ETL can be successfully printed onto a PVK:tBuCzDBA layer using the ink #2, as illustrated in Figure 3d.

### 3.2. OLED Devices

A standard multilayer OLED structure (anode/hole injection layer/electron blocking layer/EML/ETL/cathode corresponding to ITO/PEDOT:PSS/PVK/PVK:tBuCzDBA/TPBi/LiF (lithium fluoride)/Al (aluminum)) was fabricated by printing the ETL (TPBi) using solvent mixture #2 at varying solute concentrations, in order to evaluate device performance as a function of the film thickness. Specifically, three concentrations (2, 2.5, and 4 mg/mL) were tested in a MeOH:Et_2_O (8:2) mixture. The resulting film thicknesses, measured by profilometry, were 33 ± 6 nm, 48 ± 4 nm, and 99 ± 8 nm, respectively.

For comparison, and using the same basic OLED structure described above, two additional reference devices were fabricated: (a) one incorporating an ETL obtained by spin coating a TPBi solution in ethanol at a concentration of 5 mg/mL (resulting in a thickness of approximately 45 ± 6 nm); and (b) one employing an optimized thermally evaporated TPBi layer with a thickness of 50 ± 5 nm. The results are presented in Figure 4, which shows the luminance and current efficiencies of the fabricated devices as a function of the current density, allowing for performance comparison under the same injected charge conditions (Figure 4a,b). The corresponding current density–voltage curves of the best-performing OLEDs are also provided (Figure 4c). A comparative table is reported in the Appendix A. As for the devices with printed ETLs, the graphs in Figure 4 clearly show that the TPBi concentration of 2.5 mg/mL yields the best performance in terms of both luminance and current efficiency. In particular, the related curves exhibit significantly lower operating current densities compared to those of the other printed concentrations. The device with the optimal printed ETL reached a maximum current efficiency of 11.1 cd/A at 6.3 mA/cm^2^, corresponding to a luminance of approximately 700 cd/m^2^. The highest luminance recorded for this device was 3700 cd/m^2^.

For comparison, the device with a spin-coated ETL reached a maximum current efficiency of 7 cd/A and a peak luminance of approximately 8700 cd/m^2^. The OLED incorporating a thermally evaporated ETL exhibited significantly higher performance, with a current efficiency of about 24 cd/A and a luminance of 22,000 cd/m^2^. As is well known, thermal evaporation enables the deposition of highly pure films with low surface roughness and improved interface quality with the underlying layers. These characteristics favor more efficient interactions between adjacent layers, increasing the likelihood of radiative recombination of charge carriers while minimizing quenching effects due to morphological defects.

Notably, high surface roughness can promote non-radiative recombination—especially in thinner regions of the layer—by facilitating electron–hole quenching without photon emission. This phenomenon becomes especially evident when using TPBi inks at low concentrations. On the other hand, excessively high TPBi concentrations may result in overly thick layers that inhibit charge recombination, yet can improve light outcoupling due to enhanced optical confinement.

Despite these competing effects, the performance achieved by OLEDs incorporating an optimized printed ETL ranks among the best reported for fully solution-processed organic multilayer devices.

## 4. Conclusions

In the pursuit of fully printed OLEDs, a major challenge lies in developing ink formulations that ensure high-quality film deposition without compromising the integrity of the underlying layer. In this context, we report on the optimization of a TPBi-based ink for ETL printing on an emissive layer consisting of a PVK:tBuCzDBA host–guest system.

A TPBi ink based on a MeOH:Et_2_O (8:2) was identified as the most suitable for printing the ETL, effectively addressing a key bottleneck of bottom-anode OLED fabrication. Furthermore, printed ETLs of varying thicknesses were tested in OLED devices to assess their performance. Notably, the optoelectronic properties of the best-performing printed ETL device are even better than those of a comparable device with a spin-coated ETL of similar thickness, an encouraging result for the realization of fully solution-processed and fully printed OLED architectures.

## Figures and Tables

**Figure 1 materials-18-03231-f001:**
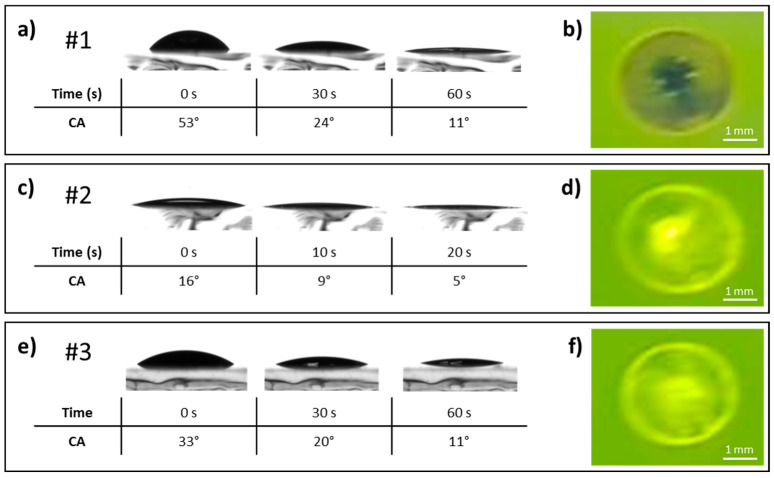
(**Left**) Contact angles and drop drying dynamics on ITO/PEDOT:PSS/PVK/tBuCzDBAof ink #1 (**a**), #2 (**c**), #3 (**e**). (**Right**) Images observed under a UV lamp irradiation (365 nm) concerning the effect on ITO/PEDOT:PSS/PVK/tBuCzDBA layer of the solvent mixture only of inks #1 (**b**), #2 (**d**), #3 (**f**).

**Figure 2 materials-18-03231-f002:**
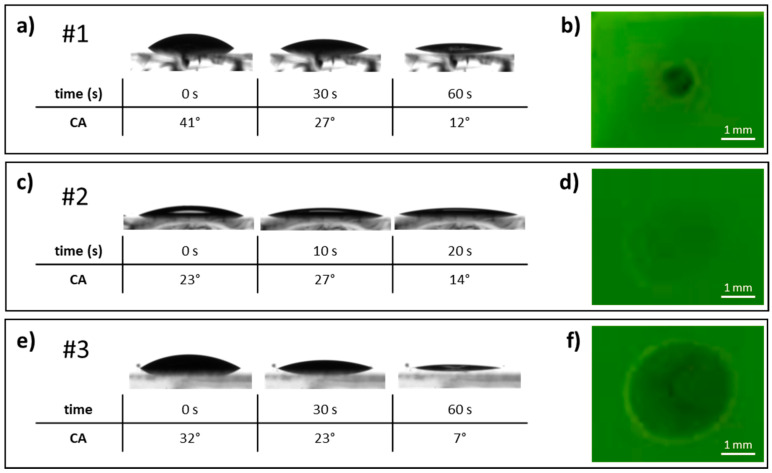
(**Left**) Contact angles and drop drying dynamics on ITO/PEDOT:PSS/PVK/PVK:tBuCzDBA of ink #1 (**a**), #2 (**c**), #3 (**e**). (**Right**) Images observed under a UV lamp irradiation (365 nm) concerning the effect on ITO/PEDOT:PSS/PVK/PVK:tBuCzDBA of the solvent mixture only of inks #1 (**b**), #2 (**d**), #3 (**f**).

**Figure 3 materials-18-03231-f003:**
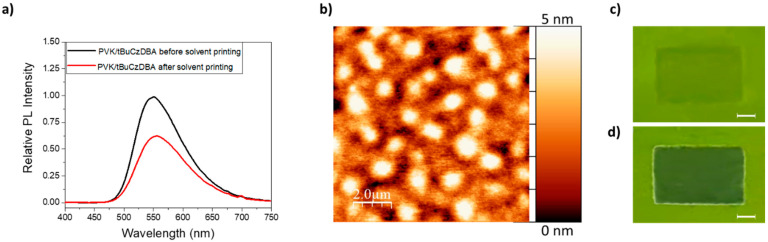
(**a**) PL spectrum of ITO/PEDOT:PSS/PVK/PVK:tBuCzDBA stack before and after printing the solvent mixture #2; (**b**) Atomic Force Microscopy (AFM) image of printed TPBi, dissolved in the solvent mixture #2, on top of ITO/PEDOT:PSS/PVK/PVK:tBuCzDBA; (**c**) image, under a UV lamp irradiation, of the inkjet-printed area (about 3 × 5 mm^2^) of the solvent mixture #2 (scale bar 1 mm); (**d**) image of the printed TPBi dissolved in the solvent mixture #2, according to the previous rectangular shape (scale bar 1 mm).

**Figure 4 materials-18-03231-f004:**
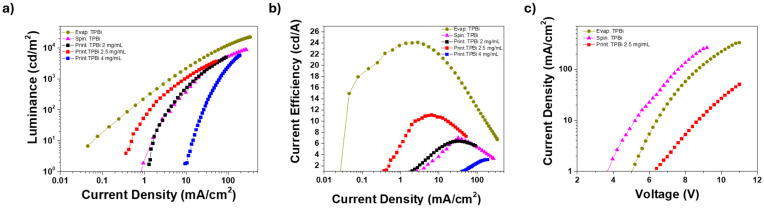
(**a**) Luminance vs. current density; (**b**) current efficiency vs. current density of the fabricated devices; (**c**) current density vs. voltage of the best OLEDs with evaporated, spin-coated, and printed electron transport layer.

**Table 1 materials-18-03231-t001:** Ink composition and surface tension values.

Ink	Solvent Composition (Volume Ratio)		Surface Tension (mN/m)
	MeOH	Et_2_O	
#1	10		25.5 ± 1.1
#2	8	2	23.1 ± 0.8
#3	6	4	22.1 ± 0.5

## Data Availability

The original contributions presented in this study are included in the article/Appendix A. Further inquiries can be directed to the corresponding authors.

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
