# Peer review of "Optimization of Electron Transport Layer Inkjet Printing Towards Fully Solution-Processable OLEDs"

_materials, 2025, doi:10.3390/ma18143231_

Round 1
Reviewer 1 Report
Comments and Suggestions for Authors This study presents a novel TPBi - based ink formulation for ETL printing on tBuCzDBA - composed emitting layers, achieving the expected composite effect via a simple solvent method, which offers valuable insights and guidance for future research on all - solution - processed OLED devices. However, several issues necessitate further elaboration: 1. The authors should provide a detailed explanation of the working principle of the inkjet printing equipment utilized in this work. 2. Based on the article, the reviewer assumes that the device employed by the authors is likely based on piezoelectric inkjet technology. The optical images in Figures 1 and 2 clearly reveal the distinctive "coffee ring" phenomenon. The authors are encouraged to emphasize the inhibitory effect of the developed ink formulation on the "coffee ring". 3. In Figure 3, the AFM image at a scale of 10 μm shows obvious and regularly distributed high points with a diameter of approximately 300 nm. The authors attribute this to the enrichment of TPBI, but further evidence is required to support this claim. Additionally, the reviewer questions whether the enrichment of TPBI implies an uneven dispersion of TPBI in the formulation. 4. The authors should supplement the characterization of the ETL layer with single - carrier testing to strengthen the study.Author Response
REVIEWER 1
This study presents a novel TPBi - based ink formulation for ETL printing on tBuCzDBA - composed emitting layers, achieving the expected composite effect via a simple solvent method, which offers valuable insights and guidance for future research on all - solution - processed OLED devices. However, several issues necessitate further elaboration:
- The authors should provide a detailed explanation of the working principle of the inkjet printing equipment utilized in this work.
Response: We thank the reviewer for the insightful comment. The printing system is based on an electromagnetically actuated microdispenser equipped with a 100 µm nozzle. The ejection mechanism relies on the vertical movement of a microneedle (or plunger) housed inside a cylindrical chamber, filled with the ink formulation. This microneedle is controlled by a precision electromagnetic actuator, which, upon receiving an electrical pulse, rapidly displaces the needle, generating a localized pressure increase that ejects a single droplet from the nozzle. The actuation follows a drop-on-demand (DoD) mode, allowing precise control over droplet size and deposition frequency. The nozzle never comes into contact with the substrate, ensuring a non-contact deposition process.
Unlike conventional piezoelectric inkjet systems, our needle-type nozzle imposes less stringent constraints on ink viscosity, allowing for the reliable jetting of formulations with a broader range of rheological properties. Although the achievable resolution is lower, this setup proves to be more versatile, particularly in our case, where the printed patterns consist of relatively large, rectangular areas without fine features. Droplet formation is optimized by adjusting the actuation parameters (pulse amplitude, duration, and frequency), the nozzle-substrate distance, the backpressure and the environmental conditions.
We have included this description in the manuscript for clarity and completeness.
- Based on the article, the reviewer assumes that the device employed by the authors is likely based on piezoelectric inkjet technology. The optical images in Figures 1 and 2 clearly reveal the distinctive "coffee ring" phenomenon. The authors are encouraged to emphasize the inhibitory effect of the developed ink formulation on the "coffee ring".
Response: The optical images in Figure 1 and Figure 2 show the effect of the solvent or mixture of solvents on the two different types of emissive layer (two different OLED stacks). Therefore, the coffee ring is due to the redissolution of the emissive layer caused by the solvent or mixture of solvents used. As can be seen in Figure 2d, no damage is observed when the redissolution is limited by selecting the emissive layer composition and solvent mixture. We have edited the text slightly and added some comments to clarify the results reported in the images more clearly.
- In Figure 3, the AFM image at a scale of 10 μm shows obvious and regularly distributed high points with a diameter of approximately 300 nm. The authors attribute this to the enrichment of TPBI, but further evidence is required to support this claim. Additionally, the reviewer questions whether the enrichment of TPBI implies an uneven dispersion of TPBI in the formulation.
Response: Thank you for your point of view. Figure 3c shows minimal degradation of the emissive layer upon solvent printing. Subsequently, the addition of TPBi in the printing solution leads to a variation in the color of the multilayer, which can thus be attributed solely to the dispersion of TPBi. This variation indicates that the thickness increase is due to a top TPBi layer, which is overall uniform. Indeed, the enrichment zone reported in the AFM picture (Figure 3b) are only a few nanometers high (see the color scale bar)—small compared to the total thickness of the layer.
- The authors should supplement the characterization of the ETL layer with single - carrier testing to strengthen the study.
Response: We thank the reviewer for this valuable and insightful suggestion. We agree that single-carrier devices can provide useful information on charge transport and material performance.
However, in our specific case, fabricating single-carrier devices would require significant modifications to the device architecture, such as altering the electrodes or removing/replacing intermediate layers (e.g., the emissive layer). Since our work focuses on the inkjet printing of the ETL directly onto a solution-processed and relatively fragile emissive layer, such modifications would introduce structural and interfacial changes that could interfere with the ink deposition process itself. These alterations may result in ETL film formation that is not representative of the actual OLED configuration and thus could compromise the relevance of the analysis.
Moreover, the deposition behaviour of the inkjet-printed ETL is highly sensitive to the nature and properties of the underlying layers (e.g., wettability, surface energy, solubility). Using a simplified single-carrier structure would not only affect these properties but could also invalidate key aspects of our ink optimization strategy.
For these reasons, instead of performing single-carrier tests, we chose to optimize the ETL ink through a combination of surface tension and contact angle studies and to evaluate the corresponding printed layer by photoluminescence, and AFM morphological analysis. These methods allowed us to evaluate the quality of the deposited ETL and the compatibility of the corresponding ink with the underlying layer under realistic fabrication conditions.
Nonetheless, we recognize the importance of single-carrier testing as a complementary characterization technique and will consider its integration in future studies, particularly for a more detailed investigation of charge transport in isolated layers.
Reviewer 2 Report
Comments and Suggestions for Authors
The authors describe the steps taken to manufacture printed OLEDs. They screened a number of solvents in order to print an ETL layer on an EML layer.
The results obtained are very interesting and necessary to be able to fabricate larger OLED surfaces.
Lines 54-55: the authors mention that the deposition of the final ETL layer is critical. It is no more critical than the others, especially if everything is printed.
When the authors test the solvents to see the impact of the solvent on the emissive layer, experimental information is missing. How did they deposit the emissive layer, and on what type of substrate?
They measured contact angles with different solvent compositions; with mixture 2, they took pictures at 10 and 20 seconds because I assume that evaporation is faster than without ether. It is not very logical that for mixture 3, they were again able to take a picture at 60 seconds.
Fig. 3: It would have been interesting to measure the optical density and determine the amount of layer that could be dissolved in addition to the photoluminescence.
Line 207: Why did they do the spin coating in ethanol since the printing is done in methanol ether? Or, since the results seem good, why not add ethanol as a printing solvent?
Figure S2 could be included in the main article. This allows us to see when the OLEDs light up.
It would be good to supplement the SI with the results obtained as indicated in the text (page 7).
Author Response
REVIEWER 2
The authors describe the steps taken to manufacture printed OLEDs. They screened a number of solvents in order to print an ETL layer on an EML layer.
The results obtained are very interesting and necessary to be able to fabricate larger OLED surfaces.
1. Lines 54-55: the authors mention that the deposition of the final ETL layer is critical. It is no more critical than the others, especially if everything is printed.
Response: We agree with the reviewer that the printing process for all OLED layers could be critical. However, the statement on lines 54-55 specifically refers to a typical bottom anode OLED structure, which could be arranged as follows: anode/HIL/EBL/EML/ETL/cathode. The anode that is typically used is ITO, followed by PEDOT:PSS and other polymers, such as PVK. For these three layers, issues of solvent orthogonality are usually limited: ITO is a layer obtained by vacuum techniques and is insoluble in solvents; PEDOT:PSS and PVK are almost insoluble after thermal annealing. Printing on the emissive layer is more challenging as it often contains small molecules, and the solvents that dissolve the emissive layer can also the ones that can be used to dissolve the ETL. Therefore, printing on top of organic layers composed of small molecules can be more challenging than printing the PEDOT:PSS, PVK or active layers, since the underlying layers are usually resistant to the solvents used in the printing process. We have tried to clarify this concept further in the text.
2. When the authors test the solvents to see the impact of the solvent on the emissive layer, experimental information is missing. How did they deposit the emissive layer, and on what type of substrate?
Response: Information relating to the preparation of the emissive layer for the 'solvent test' is reported in sections 4.2 and 4.3. The tests were carried out on two types of OLED stack with the following structures: ITO/PEDOT:PSS/PVK/tBuCzDBA and ITO/PEDOT:PSS/PVK/PVK:tBuCzDBA. All details relating to the solutions used and the deposition techniques employed are provided. We have modified the text to specify the substrate used for the tests more clearly and the changes are highlighted.
3. They measured contact angles with different solvent compositions; with mixture 2, they took pictures at 10 and 20 seconds because I assume that evaporation is faster than without ether. It is not very logical that for mixture 3, they were again able to take a picture at 60 seconds.
Response: The contact angles were monitored over a period of 90 seconds, with significant frames reported in the images. For ink #1, the initial contact angle is high and the ink completely wets the substrate within 60 s. For ink #2, the initial contact angle is much lower, resulting in faster wettability dynamic, with the contact angle almost reaching zero within 20 s. For ink #3, the opposite trend to that expected was observed, with the contact angle at time zero being higher than for ink #2, as also highlighted in the text. As the contact angle increases, the time required for good wettability also increases, so it was deemed appropriate to report the contact angle values ​​at 60 s.
The reported contact angles and the relative times are therefore related to the initial contact angle and wettability dynamics. Evaporation/drying time is affected by the boiling point of the solvent used, as well as by the wetting behaviour. When the contact angle is low, the contact area between the liquid and the air is greater, resulting in faster evaporation/drying.
Therefore, the addition of diethyl ether certainly influences the evaporation process; however, in this case, the reported times are more related to the wetting behaviour than to the quantity of diethyl ether added to the inks. We have added comments to the text to clarify some details and the changes are highlighted.
4. Fig. 3: It would have been interesting to measure the optical density and determine the amount of layer that could be dissolved in addition to the photoluminescence.
Response: We thank the reviewer for this suggestion. Unfortunately, we are currently unable to perform optical density measurements in a short time, as our spectrophotometer is out of service. However, we carried out thickness measurements before and after the solvent printing process to determine the variation in thickness due to the material dissolution and accumulation at the edges. After several measurements we estimated a reduction of about 10-15%. We have added this information in the text.
5. Line 207: Why did they do the spin coating in ethanol since the printing is done in methanol ether? Or, since the results seem good, why not add ethanol as a printing solvent?
Response: The goal of using spin-coated TPBi is to provide a reference device for comparing the performance of fully spin-coated OLEDs with those incorporating a printed ETL. To obtain a spin-coated ETL of a similar thickness, we used a recipe that was previously developed in our laboratories, where ethanol proved to be effective for the spin-coating process. As a general remark, we emphasize that the optimization achieved with one deposition technique cannot be directly transferred to another, as certain parameters may differ and significantly affect the outcome. In our case, preliminary tests using ethanol for ETL printing revealed excessive damage to the emissive layer, indicating that ethanol is not suitable for printing the subsequent layer. Here, the crucial factor that makes ethanol compatible with spin-coating but not with printing lies in the solvent–film contact time, which is considerably shorter in spin-coating, thus reducing the risk of damage or redissolution of the emissive layer. Concerning printing, therefore, since methanol proved effective in dissolving TPBi while also limiting damage to the underlying layer, we optimized the deposition process using this solvent.
6. Figure S2 could be included in the main article. This allows us to see when the OLEDs light up.
It would be good to supplement the SI with the results obtained as indicated in the text (page 7).
Response: We thank the reviewer for this suggestion. We moved Figure S2 in the main text (Figure 4) and we added a comparative table for OLED performance in SI.
Reviewer 3 Report
Comments and Suggestions for Authors
Please review the abbreviations you have entered carefully in the text again. Some abbreviations are entered several times at once. Some are used without being explained beforehand.
There are some minor grammatical errors.
Given pictures are not the same size in Fig.3 and 4.
Please more explain why did You decide to use methanol? You mentioned that you decided it because it was was intermediate solvent. What could happen if you decided to choose n-butanol or ethyl acetate?
2.2 OLED devices: you wrote that TPBi is ETL, can you explain about another layers? What layers are PEDOT:PSS, LiF and so on?
It is quite interesting why did You decide to use such ratio of solvents (10:0; 8:2 and 6:4)? What if you tried using 9:1 or 7:3?
Author Response
REVIEWER 3
1. Please review the abbreviations you have entered carefully in the text again. Some abbreviations are entered several times at once. Some are used without being explained beforehand.
Response: We thank the reviewer for this advice. We reviewed the abbreviations in the manuscript.
2. There are some minor grammatical errors.
Response: We thank the reviewer for this suggestion. We have double-checked the manuscript and corrected grammatical errors. All changes are highlighted.
3. Given pictures are not the same size in Fig.3 and 4.
Response: We checked and modified figures size.
4. Please more explain why did You decide to use methanol? You mentioned that you decided it because it was intermediate solvent. What could happen if you decided to choose n-butanol or ethyl acetate?
Response: Another intermediate solvent evaluated was ethanol; however, it was later discarded because of its adverse impact on the active layers. n-Butanol was not considered because previous experience with out printing system has shown that solvents with a high boiling point lead to a considerable increase in drying time and coffee-ring formation.
Considering the boiling temperature and surface tension, ethyl acetate could be a good starting solvent like methanol, however its ability to redissolve the emissive layers has not yet been tested.
Further tests could be carried out in the future, but we believe that this work's objective of identifying a suitable ink for printing TPBi on a tBuCzDBA-based emissive layer has been achieved.
5. Paragraph 2.2 OLED devices: you wrote that TPBi is ETL, can you explain about another layers? What layers are PEDOT:PSS, LiF and so on?
Response: We added details related to the used OLED architecture.
6. It is quite interesting why did You decide to use such ratio of solvents (10:0; 8:2 and 6:4)? What if you tried using 9:1 or 7:3?
Response: The solvent ratios examined in this study refer to the progressive replacement of methanol with increasing amounts of diethyl ether. While additional intermediate ratios could certainly have been prepared and tested, we expected that their behavior would simply fall between that of the ratios already investigated.
The 8:2 methanol:diethyl ether ratio yielded satisfactory results, ensuring uniform coverage of the active layer and preventing redissolution of the emissive material. Moreover, unlike the 6:4 mixture—where TPBi tends to precipitate over time—the 8:2 solution remained stable. For these reasons, we selected the 8:2 ratio for OLED device fabrication and did not pursue further optimization of other ratios. Further experiments could be carried out in the future, but the results reported here satisfy the objective of printing a TPBi layer on a tBuCzDBA-based active layer.
Reviewer 4 Report
Comments and Suggestions for Authors
- Indicate what is the quality of all solvents used, purity and if they were dried before or used without any purification process.
- For milliliters used the correct acronymous mL instead of ml.
Author Response
REVIEWER 4
- Indicate what is the quality of all solvents used, purity and if they were dried before or used without any purification process.
Response: We thank the reviewer for the helpful remark. We have now included information about the purity of the solvents in the Materials and Methods section, specifying that they were used as received, without any further purification.
- For milliliters used the correct acronymous mL instead of ml.
Response: We corrected the acronymous in the revised manuscript.